# MAEA: Multimodal Attribution for Embodied AI

**Vidhi Jain    Jayant Sravan Tamarapalli\*    Sahiti Yerramilli\*    Yonatan Bisk**
Carnegie Mellon University
{vidhij,jtamarap,syerrami,ybisk}@andrew.cmu.edu

## Abstract

Understanding multimodal perception for embodied AI is an open question because such inputs may contain highly complementary as well as redundant information for the task. A relevant direction for multimodal policies is understanding the global trends of each modality at the fusion layer. To this end, we disentangle the attributions for visual, language, and previous action inputs across different policies trained on the ALFRED dataset. Attribution analysis can be utilized to rank and group the failure scenarios, investigate modeling and dataset biases, and critically analyze multimodal EAI policies for robustness and user trust before deployment. We present MAEA, a framework to compute global attributions per modality of any differentiable policy. In addition, we show how attributions enable lower-level behavior analysis in EAI policies for language and visual attributions.

## 1 Introduction

Embodied AI policies have achieved remarkable success in simulated 3d environments [1, 2, 3, 4] and physical robots [5, 6]. Analogous to the success of end-to-end learning for image classification [7] and language modeling [8], recent works on end-to-end embodied policies [9] attempt to solve complex everyday tasks, like 'making a cup of coffee' or 'throwing a chilled tomato slice in trash' (as shown in Fig. 1). Such policies often fail for reasons that are poorly understood. Interpreting the decision-making in end-to-end policies is important to enable trustworthy deployment and handle failure case scenarios.

Learning an embodied AI policy typically involves function $f$ mapping the observation/state $O_t$ to action $a_t$. In many environments [10, 11], the observation can just be the visual frame at current timestep $O_t = \{V_t\}$. However, complex tasks can hardly be solved with just a single type of observation. In Atari [12], the policy operates on last 4 visual frames: $O_t = \{V_i\}_{t-3}^{t}$. For long horizon tasks in a large sparse maze environment [13], inputs may include the previous action: $O_t = \{V_t, a_{t-1}\}$, as it will be likely that the following action is to 'move forward' if the previous action was the same. Real-world robotics navigation relies on visual frame and proprioception [6] to successfully navigate undetected obstacles. In this work, we consider a mobile robot that takes natural language instructions and interacts with objects in simulated household environments [9]. Such a robot needs to predict the sequence of actions that would complete the task, given the previous action $a_{t-1}$, natural language instructions $L$, and ego-centric vision $V_t$, that is the observation $O_t = \{V_t, a_{t-1}, L\}$. While many attempts have been made to train

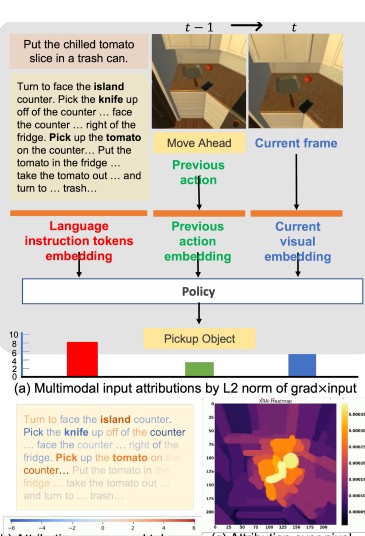

Figure 1: Attribution analysis of multimodal EAI policy. (a) (left to right) shows attribution for language, previous action and visual frame. (b) **Words**: red is positive attribution, blue is negative, grey is neutral; (c) **Pixels**: yellow is high attribution, purple is low.

\* equal contribution

Table 1: Policies trained on ALFRED Dataset and their architectures for each modality

| Policies | Visual | Language | Fusion |
|---|---|---|---|
| Baseline [9] | Frozen ResNet-18 | Learned Embedding, Bi-LSTM | LSTM |
| MOCA [14] | Frozen ResNet-18 + Dynamic Filters | Learned Embedding, Bi-LSTM | LSTM with residual connection |
| ET [15] | Frozen ResNet-50 | Learned Embedding, Transformer | Transformer Encoder |
| HiTUT [31] | Frozen MaskRCNN | Learned Embedding, FC, LN | Transformer Encoder |

policies for this task [9, 14, 15, 16], the existing best performance is far below that of an average human [9]. To this end, we investigate how multimodal a policy is, in terms of attribution given to visual, language, and previous actions. Aggregated analysis per modality provides insights into the effect of modeling choices, such as at what layer the fusion happens, what sub-network is used to process each modality separately, and how the representation per modality affects the fusion process. Further, attributions provide an introspection technique to rank the contributions from each modality in a decision, as well as analyze modeling and dataset biases. Our main contribution is to propose Multimodal Attribution for Embodied AI, MAEA, for (a) global analysis in terms of the percentage of average attribution per modality, as well as (b) modality-specific local case studies. Our work does not intend to comment on the kind of attributions useful for understanding multimodal policy but only provides a tool to better understand the modality attributions in any model architecture setting.

## 2   Related Work

**Interpretability and explainability**   Recent work in multimodal explainability in autonomous vehicles [17] uses symbolic explanations to debug and process outputs out of sub-components. In contrast, we address the challenge of post-hoc multimodal interpretability for any existing end-to-end trained differentiable policies. GRAD $\odot$ INPUT [18], a simple and modality-agnostic attribution that works on par with recent methods [19]. We use this method to compute multimodal attribution at inputs to the fusion layer to weigh how each modality contributes to the decision-making. While GRAD $\odot$ INPUT is a modality-agnostic starting point for attributions, it is not easy to understand, especially for images. Among recent works to improve visual attribution [20, 21, 22, 23, 24, 25, 26], we use XRAI [26] for vision-specific analysis as it produces visually intuitive attributions by relying on regions, not individual pixels.

**Language-driven task benchmarks**   There are many benchmarks to study an agent's ability to follow natural language instructions [9, 27, 28, 29]. ALFRED [9] serves as a suitable testbed for this analysis as these tasks require both high reasoning for navigation and manipulation. ALFRED dataset provides visual demonstrations collected through PDDL planning in 3D Unity household environments and natural language description of the high-level goal and low-level instructions annotated by MTurkers. The benchmarks provide evaluation metrics for the overall task goal completion success rate (SR) and those weighted by the expert's path length (PLWSR) and have reported a huge gap in the performance of learning algorithms and humans at these tasks.

**End-to-end Learned Policies**   We investigate the end-to-end learned policies for the task, such that, the gradient can be attributed at a task level. While we do not discuss modular yet differentiable policies like [16] [30], tying the gradient across multiple modular learned components is a direction for future work. In our work, we consider the checkpoints of policies trained on the ALFRED dataset. Broadly, these policies are of two types: (a) sequence-to-sequence models, that are, the one proposed with ALFRED dataset (Baseline) [9] and Modular Object-Centric Approach (MOCA) [14], (b) transformer-based models, that are Episodic Transformers (ET) [15], and Hierarchical Tasks via Unified Transformers (HiTUT) [31]. Refer Table 1 to compare architectural details [1].

## 3   Approach

Gradients are the general way of discussing the coefficient of the importance of a particular feature in deciding the output. Using only weights or gradients as an attribution assumes that the values of $x_1$

---
[1] Previous action is modeled with learned embedding look-up in all these policies.

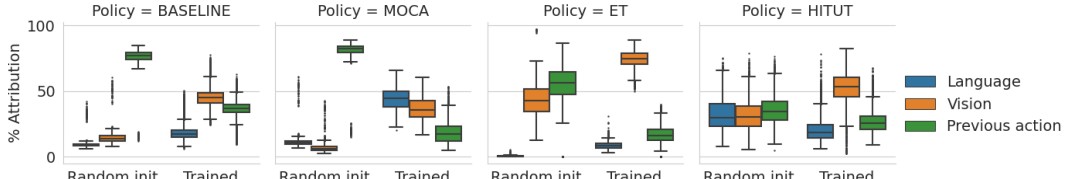

Figure 2: % Multimodal Attribution for policies *randomly initialized* and *trained* on ALFRED dataset. The performance of the policies improves from left to right. Trained checkpoints are evaluated with 100 trajectories sampled from validation-seen trajectories. Policies are randomly initialized for 5 seeds each and evaluated over 10 validation-seen trajectories. Note that, in this case, performance improves as the skewness in the attributions prior to training decreases.

and $x_2$ are of the same order, like image pixels. However, this doesn't hold in the case of multimodal policies when $x_1$ is an image embedding and $x_2$ is a language embedding. *Element-wise product of gradient into input*, also known as GRAD $\odot$ INPUT [18], provides global importance about the input feature in the dimensionality of the input feature itself. We compute the attributions per modality with respect to the predicted action label output. The differentiable policy $f : (V_t, a_{t-1}, L) \rightarrow a_t$ takes input as current visual frame $V_t$, previous action label embedding $a_{t-1}$, and the language instruction $L$. At the penultimate fusion layer, an intermediate representation is typically a vector or matrix. The attribution for each modality can be computed by GRAD $\odot$ INPUT, where input is the feed-forward features computed at the penultimate fusion layer. Let the policy neural network be $f$ that outputs a softmax distribution over the action to be taken. The input feature attribution $\alpha_i$ for $i^{th}$ dimension of vector $\mathbf{x}$ for the most likely predicted action is computed as $\alpha_i = \frac{\partial(\max f(x))}{\partial x_i} \odot x_i$. For each modality represented as a vector, we need to pool the attribution per dimension to compute a scalar value attribution. Here are the implications of different pooling approaches:

- $L^\infty$     gives the maximum magnitude value, independent of the dim $d$ (same as $L^\infty/d$).
- $L^1$       provides the sum of all attribution magnitudes.
- $L^1/d$   is same as $L^1$, but invariant of the dim $d$.
- $L^2$       diminishes the majority insignificant but non-zero attributions.
- $L^2/d$   has same impact of $L^2$ but with undesirable scaling by dim $d$.

To compute modality-specific attribution for latent vectors, $L^2$ is suitable to include the attributions from every dimension in the vector (unlike $L^\infty$) and reduce the impact of insignificant close to zero attributions (unlike $L^1$). We do not consider $L^1/d$ or $L^2/d$ as the attribution of modality should depend on the number of latent features allocated to highlight modeling biases. Note that for global attribution, we treat the feature extraction in each modality as a black box as we capture attributions at the penultimate fusion layer.

## 4   Global Attribution Analysis

To understand how the multimodal attribution at the fusion layer is, we analyze the GRAD $\odot$ INPUT with L2 norm as pooling to compute attributions. In Fig. 2, we show the percentage of attribution given to language, vision, and previous action before and after training by different policies on ALFRED dataset. The attributions before training represent the implicit bias in the architecture of the model since it has not seen any data yet. The attributions after training, given the ones before training, represent the bias that is introduced by the data. This decoupling of interpretability of the biases introduced by model architecture and training data provide better equips the user to associate the bias between the two aspects. See Appendix C.

Baseline and HiTUT have a balanced attribution after training over all three modalities, with a preference for visual features. While MOCA prefers language slightly over visual and previous action, ET strongly prefers visual features over previous action and language. At initialization, Baseline and MOCA have the majority of the % attribution on the previous action. ET model starts with the majority of the attribution over previous action and visual frame, and very little focus on the language. HiTUT has a balanced attribution before training, which does not inherently induce modality bias. More details in Appendix B.

# 5 Modality-specific Attribution Analysis

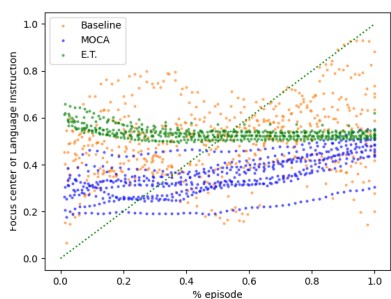

**Does language attribution change depending on the steps taken in the episode?** For language attribution, we compute the gradient of the predicted output with respect to embeddings of the word tokens in the language instruction, $M := \texttt{Embedding}(\texttt{Tokenize}(L))$, where $M \in \mathbb{R}^{n \times d}$. We compute the attribution for the matrix $M$ by $\alpha_M := \frac{\partial f}{\partial M} \odot M$. As word embedding is a $d$-dimensional vector, the gradient is also computed with respect to each feature dimension. For pooling, we use the maximum absolute value as the attribution as $\alpha_w := \max \texttt{ abs}(\alpha_M[i]), \forall i \in n$. To see how the focus center of the attribution for language instructions shifts in terms of the percentage of episode completion (Fig. 3[2]), we compute the word-index weighted attri-

Figure 3: Focus center of Language attributions per episode length: Baseline [9] (yellow), MOCA [14] (purple) and ET [15] (green) on validation seen dataset.

bution over all the word tokens in instruction per trajectory $c_{raw} = [(i+1) * \text{abs}(\alpha_w)] \forall i \in n$. To compare across variable length trajectories, we normalize it as $c_{scaled} = \sum c_{raw}/(\sum \text{abs}(\alpha_w) * n)$. In the ideal case, the focus center of attribution on instruction should increase as the episode nears completion. MOCA (blue) follows a linearly increasing trend in Fig. 3 and has high language attribution in Fig. 2. Baseline and ET do not show such a trend, which aligns with their low attribution to language in Fig. 2.

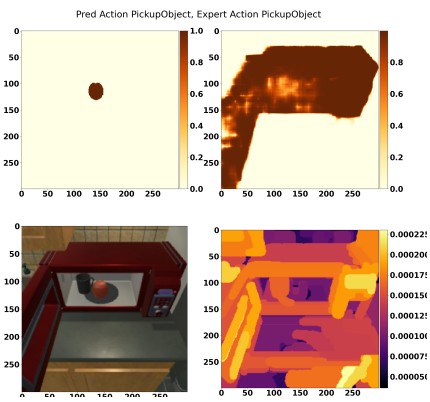

Figure 4: Visual attributions for an interactive action with an object can indicate the focus of the baseline policy. Top-Left: Expert's interaction mask, Top-Right: Predicted interaction mask, Bottom-Left: visual observation in RGB, Bottom-Right: XRAI attributions.

**Does visual attribution align with the predicted interaction mask?** We analyze how visual attributions for interact action (PickObject) can indicate the focus of the interaction mask. We calculate the visual attribution with XRAI [26] as shown in Fig. 4 for Baseline policy. While the action label is predicted correctly by the policy, the intersection over the union of the mask is small. The simple visualization of the interaction mask reveals that a lot of objects (microwave, mug, and tomato) are selected. With XRAI attribution, we can qualitatively analyze the regions in terms of high (eg. parts of the microwave) and low (eg. tomato) attribution. Visual attribution can be used to analyze failure cases where the policy predicts the correct interact action but the wrong action mask. While the action mask is a heatmap on the visual image to determine the IOU with the groundtruth object mask, XRAI attribution provides some insight based on the regions, and not just pixels, about which parts the policy is focusing on while predicting the action label. Note that this is an interpretability and analysis tool to debug failure cases, and not a complete solution for mitigating errors in predicting interaction action masks.

# 6 Conclusion

In this work, we draw the community's attention to attribution analysis for interpreting multimodal policies. We provide the framework MAEA for attribution analysis to gain insights into multimodal embodied AI policies. We compare seq2seq (baseline, MOCA) and transformer-based (ET, HiTUT) policies trained on the ALFRED dataset and highlight the modality biases in these models. We also analyze how the focus center in language instructions moves as the episode progresses, and discuss how visual attributions can be used for analyzing successful/unsuccessful action predictions. Note while we use this gradient-based attribution, the ideas of multimodal attribution can be generally applicable to other kinds of attributions. This technique could also be used to better understand the correlation between certain types of biases and failure cases in certain in-distribution as well as out-of-distribution scenarios.

---

[2]HiTUT is not shown in Fig. 3 because it doesn't consider the previous language instructions to make the current prediction which is different from how the other papers do it.

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

## A More Multimodal Attribution Analysis

We compare the multimodal attribution in terms of interact vs non-interact action (Fig. 5), and how they may change with respect to % episode completion (Fig. 6). We also compare the biases in attributions to the modalities because of choices in model architecture and the way training is performed, i.e., dataset and learning techniques.

### A.1 Interact vs non-interact action attributions

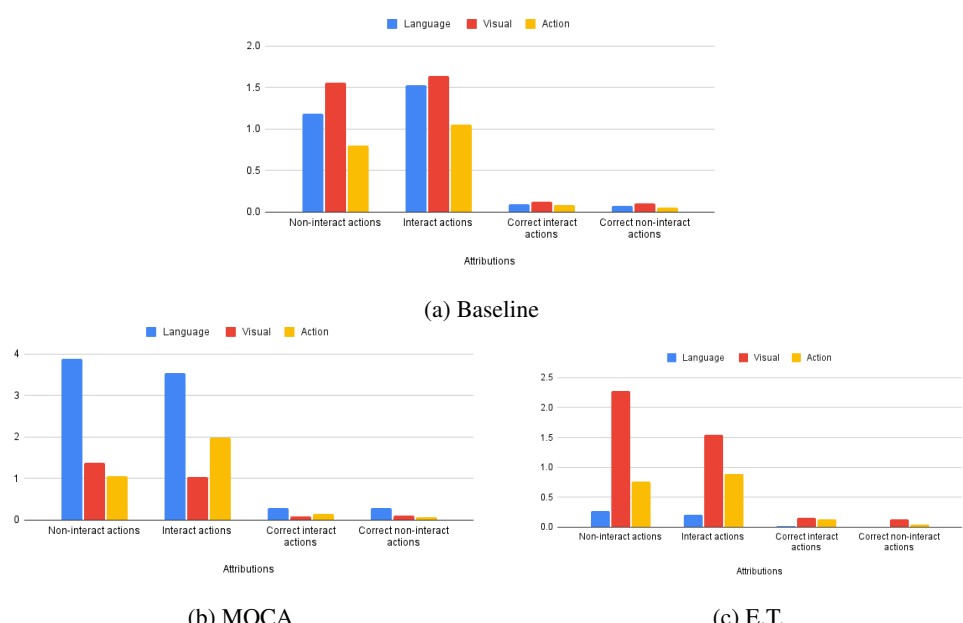

(a) Baseline

(b) MOCA

(c) E.T.

Figure 5: L2 norm of the attribution by grad ⊙ input for interact and non-interact actions.

We analyze the L2 norm of the attribution over all the possible input-output pairs in valid seen and unseen trajectories and bin them in terms of interact (like pick, place, open, close, etc.) and non-interact (like move ahead, turn left, turn right, etc) actions in Fig. 5. This is helpful to visualize how the attributions differ with (i) different types of actions taken and (ii) correct and incorrect predictions. We observe that the attribution patterns do not differ significantly in interact vs non-interact action.

### A.2 Attribution with respect to % episode completion

To compare the overall attribution over modality with respect to where the action is taken in terms of % episode length, we plot the attributions for the models trained on ALFRED task in Fig 6a, 6b, 6c and 6d to visualize how the attributions change during the episode for baseline, MOCA, ET and HiTUT respectively. In an ideal case, we would expect more attribution on visual and previous action features in the exploration phase when starting in a new environment, and more on language instruction towards the later part of the episode completion phase. But in current models, we observe that the attribution pattern remains consistent over the episode – indicating a possible need for improvement in modeling choices and training procedures.

The baseline has high attribution for visual features, especially initially and toward the end of the episode. Attribution for previous action and language also increases near episode completion steps. MOCA has high attributions over the language instructions as compared to the visual frame embedding and previous actions. Previous action attribution closely overlaps with language attribution. Visual attribution increase towards the episode completion. ET shows significantly high visual attributions that increase and then plateau towards episode completion. Previous action and language seem to have very low attribution compared to visual features. HiTUT, which is the best-performing model of all, shows sufficiently balanced attribution among all three modalities, with occasional high peaks.

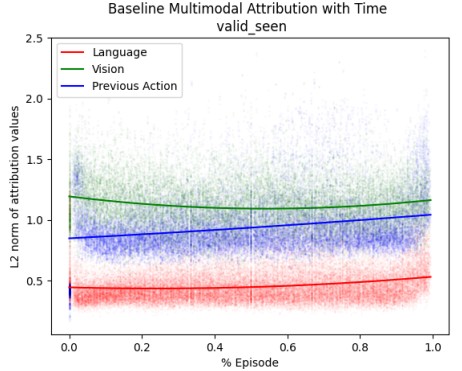

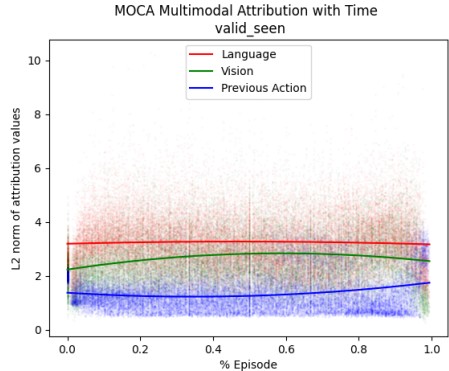

(a) Baseline [9]: Previous action attribution is higher than vision. Language is the lowest.

(b) MOCA [14]: Language attribution is the highest, especially towards the end of the trajectory.

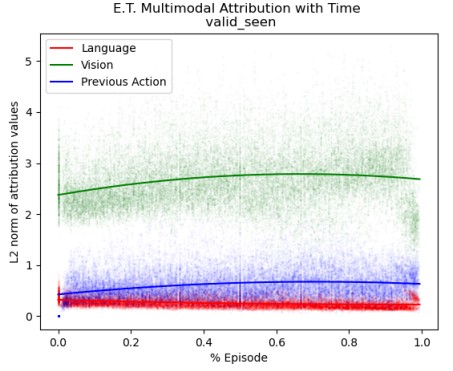

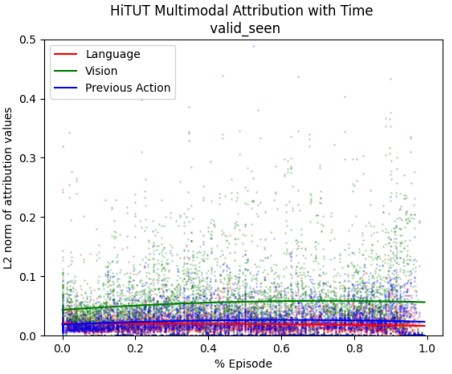

(c) E.T. [15]: Visual attributions are the highest, followed by those for the previous action taken.

(d) HiTUT [31]: Overall attributions are low. Vision is higher, more towards the end.

Figure 6: Multimodal attributions in ALFRED policies over % episode completion for seen validation.

## B   What is the ideal target to estimate multimodal attribution?

We compute the attribution of each modality/modality-specific feature with respect to the action predicted by the agent, that is $f$ is a scalar value for the most likely action. Other values for function $f$ can be other scalar values such as the loss used during training or the action taken by the expert at that particular timestep, or tensors, like the entire action space (since a policy would return logits for all actions in most cases) or the predicted interaction masks. Taking attributions with respect to the loss tells us what the agent should be looking at to better imitate the expert. But in terms of what features the agent is using to select an action gets conflated with the gradient of the loss with respect to that action. Taking attributions with respect to the expert action would result in an interpretation that would convey the attributions that would have led to the agent taking the expert action. Considering all the logits corresponding to the action choices makes sense for our purposes but there is a concern that not all model architectures allow us to backpropagate to inputs from all the possible discrete actions. Predicted interaction masks are a good indication of what the model is looking at while making an interact action. But this is a vision-biased attribution and also assumes that the model computes an interaction mask in the first place. Therefore, gradient with respect to the action taken is a better choice to interpret the policy decisions and it intuitively translates to the objective of filtering for inputs that gave rise to a certain decision.

## C   Global Attribution Experiment Setup

The attributions prior to training are used to analyze modeling biases. To get the attributions before training, we use randomly initialized policies over 5 seeds. For the trained model's attributions, we

use pretrained checkpoints. Both attributions are evaluated by a sample of 100 trajectories from 820 in validation-seen data.

We compute these attributions based on the expert trajectories to keep the analysis on the more relevant input states within the training distribution. We do not analyze the rollout trajectories because most policies have poor success rates (SR) and are often stuck in irrelevant out-of-distribution states, such as facing a wall. We take the gradients with respect to the most likely action predicted by the policy, and not the expert's chosen action.

## D    Limitations of the attribution methods

Attribution methods might be sensitive to the choice of pooling used. Here we defend our formulation, why L2 norm - because we want to compare policies that may use different dimension embeddings to represent an input. For example, policy 1 uses $d_1 = 128$ dim to represent word token embedding, policy 2 uses $d_2 = 768$ dim to represent the same. There are two factors here: first, in higher dimensions, the L1 norm gives a better distance estimate than the L2 norm. But, by the virtue of how the neural networks are initialized for stable training, the higher dimension will have a lower value per dimension. This means that the absolute attribution value in policy 1 should be scaled by its embedding size for a fair comparison to policy 2.

