# OpenReview forum: "MAEA: Multimodal Attribution for Embodied AI"
_NeurIPS.cc/2022/Workshop/TEA — TEA_

### Official Review · Reviewer_kzf8 · 2022-10-05
**Analyzing attribution for multi-modal policies**

**Rating:** 6
**Confidence:** 3

**Review:**

Interesting paper on post-hoc interpretability of end-to-end multimodal policies (where the modes here are language, vision, and last action). First we’re presented with a global attribution analysis for 4 different policy learning models. The paper then does a deep dive into each modality type (language and vision) separately to further investigate attribution. For future iterations of this work, I recommend spending more time building up the motivation for why multi-modal attribution is important in the real-world and giving some examples for how it could be helpful (basically, grounding the introduction in an example would be more compelling because the manuscript is quite abstract and procedural right now). I also would have wanted to see some more justification behind the choice for the 4 models that were analyzed. Lastly, I recommend adding some discussion on how this post-hoc attribution analysis model could be useful. In general, the results that are presented seem compelling but there isn’t enough motivation/grounding/explanation to understand why they’re important/matter.

---

### Official Review · Reviewer_Rvmk · 2022-10-16
**Should be more critical of attribution methods!**

**Rating:** 6
**Confidence:** 5

**Review:**

Summary
---
This paper adapts feature attribution methods for multimodal inputs, e.g., consisting of both text and images.

Strengths
---
The motivation of this paper - particularly as specified in the introduction - is compelling. Specifically: "Interpreting the decision-making in end-to-end policies is important to enable trustworthy deployment and handle failure case scenarios."

Weaknesses
---
Feature attribution methods are plagued by a lack of agreement on the goals and evaluation mechanisms of these tools; e.g., see Sanity checks for saliency maps by Adebayo et al, Do feature attribution methods correctly attribute features? by Zhou et al, or The Mythos of Model Interpretability by Lipton. In light of these disagreements and skepticism in the community, any work which uses or engages with feature attribution methods should adopt a critical approach. We must ask: What is the goal of the feature attribution method in use? How will we measure its success in achieving this goal?

Other comments
---
This is quite a high level comment and more for the PC than the authors. I appreciate the paper disclosing that it is in dual submission with another NeurIPs workshop. While I'm not sure what TEA's specific policy is on this, in principle I do not like the idea of submitting papers to multiple workshops.

One of the main evaluation measures is % attribution. Why is this a good measure? What are the expectations we should have, and why?

I appreciate the inclusion and discussion of several different pooling approaches and their implications for attribution.

---

### Official Review · Reviewer_4Q9k · 2022-10-20

**Rating:** 7
**Confidence:** 3

**Review:**

This paper is novel and well-written, delivering some very interesting insights into multi-modal frameworks for embodied AI. As far as I know, few researchers attempt to explore how the model pays attention to different modal attribution.

The exploration method for attribution importance, input and grad, makes sense, and the study in this paper is very inspiring: it seems that visual modality dominates the impact of the policy if it is well per-trained while previous actions seem to be less significantly participating the decision making. If no pre-trained model is provided, the conclusion seems to be the opposite. The language attention seems hard to optimise reasonably. Shall we pay more attention to visual representation and put efforts into how to better utilize language prompting?

The drawback of this paper is that there is no more insight into why the random initialization framework and the pre-trained framework have different modality attribution importance to the policy. On the other hand, it is not clear how different the final performance is between these two settings. Do we need to trade off the pre-trained models?

Overall, I think this paper is a good short paper with two spotlights. I would recommend it.

---

### Decision · Program_Chairs · 2022-10-21

**Decision:**

Accept

**Comment:**

The paper is well-written and provide interesting and important insights into interprepting the decision-making process of end-to-end multimoal policies. All the reviewers have a positive view of the paper. Please consider the comments from the reviewers in the final version.